# A unique Pd-catalysed Heck arylation as a remote trigger for cyclopropane selective ring-opening

Sukhdev Singh[1], Jeffrey Bruffaerts[1], Alexandre Vasseur[1] & Ilan Marek[1]

Combining functionalization at a distant position from a reactive site with the creation of several consecutive stereogenic centres, including the formation of a quaternary carbon stereocentre, in acyclic system represents a pinnacle in organic synthesis. Here we report the regioselective Heck arylation of terminal olefins as a distant trigger for the ring-opening of cyclopropanes. This Pd-catalysed unfolding of the strained cycle, driving force of the chain-walking process, remarkably proved its efficiency and versatility, as the reaction proceeded regardless of the molecular distance between the initiation (double bond) and termination (alcohol) sites. Moreover, employing stereodefined polysubstituted cyclopropane vaults allowed to access sophisticated stereoenriched acyclic scaffolds in good yields. Conceptually, we demonstrated that merging catalytically a chain walking process with a selective C–C bond cleavage represents a powerful approach to construct linear skeleton possessing two stereogenic centres.

[1] The Mallat Family Laboratory of Organic Chemistry, Schulich Faculty of Chemistry and the Lise Meitner-Minerva Center for Computational Quantum Chemistry, Technion-Israel Institute of Technology, Haifa 32000, Israel. Correspondence and requests for materials should be addressed to I.M. (email: chilanm@tx.technion.ac.il).

I n the last few decades, the field of organic synthesis has witnessed tremendous evolutions. Traditionally, the organic chemistry community mainly focused its efforts on forming new covalent bonds at a single and generally most reactive functional group to interconvert functionalities (Fig. 1a)[1,2]. In recent years a paradigm shift occurred, deeply confronting our generally temporally linear retrosynthetic mindset to the benefit of more flexible, versatile and economical transformations. For instance in the field of organometallic chemistry, powerful strategies have been developed where specific metal-mediated reactions may be performed in the presence of other functional groups, thus drastically improving the integration of organometallic reagents in synthesis (Fig. 1b)[3,4]. Despite this impressive portfolio of synthetic strategies developed so far, it still appears challenging to functionalize at distant, unrelated and often unreactive positions from any functional group on a molecule (Fig. 1c). To address this issue, the concept of remote functionalization—where an initial interaction of a functional group leads to a selective reaction at a distal position—was introduced several decades ago[5,6]. Even though this strategy has rarely been considered in synthesis so far, recent reports have nonetheless contributed to substantially promote this powerful and singular approach[7–20]. In particular, a remote functionalization proceeding through transition metal-mediated chain-walking[21] has the significant advantage of increasing the versatility and scope of a transformation through a communicative process associating two distant functional groups independently of their molecular distance (n) (ref. 22). An initial chemoselective interaction between a metal complex and a functional group (FG[1], initiation site) may trigger the induction of a second functional group interconversion at a distant site (FG[2], termination site), assuming an efficient and rapid metal complex migration along the molecule. This strategy potentially allows the interconversion of two functional groups at positions unrelated from each other in a single reaction vessel. However, chain-walking reactions are generally limited to saturated methylene connections between the functional groups. Although stereochemical information of tertiary centres might be conserved during the migration[13], quaternary centres would prevent any metal migration process to occur until the desired termination site (Fig. 1c). To circumvent this latter problem, we hypothesized that one could develop a chain-walking/cyclopropane ring-opening/chain-walking sequence as depicted in Fig. 1d. The subsequent cyclopropane 'vault' would, therefore, serve as a readily frangible platform to install internal substitutions between two distant functional groups on a same linear hydrocarbon chain. Merging metal-mediated alkene migration[23,24] and selective C–C bond cleavage[25–28], two cutting edge methods in organic synthesis, would represent a novel approach to mould strained cyclic molecules into sophisticated acyclic molecular skeletons. Over the last few years, our laboratory has been involved in the specific development of remote activation reactions[29–32] using stoichiometric amount of low-valent zirconocene complexes. Among others, we disclosed the highly selective transformation of ω-ene cyclopropanes into acyclic bifunctionalized (E)-olefins (Fig. 1e)[33,34]. In light of this powerful one-pot entry to acyclic systems, we recently questioned the possibility of developing such sequence using a catalytic amount of a transition metal based complex enabling concomitantly C–C bond coupling, metal migration and a selective cleavage of a C–C bond.

## Results

**Reaction design**. For this purpose, one should consider that a catalytic long-range activation heavily relies on a rapid metal migration over the hydrocarbon chain and fast release of the metal from the termination site[22]. Following these requirements, we turned our attention to the Pd-catalysed Heck reactions[33] that have been previously known to trigger an efficient migration of the organometallic species over hydrocarbon chains[34]. Based on the pioneering observation of Heck who reported the Pd-catalysed arylation of 4-penten-2-ol into 5-phenyl-pentan-2-one, although in low yield[35], Larock consequently further developed this transformation with other non-allylic unsaturated alcohols[36] (Fig. 2a). Ultimately, these works led to the most recent *tour de force* when Sigman successfully developed the enantioselective version of this reaction (Fig. 2b)[13,37]. Remarkably efficient and highly selective Heck arylation reactions over unactivated di- or trisubstituted alkenes and subsequent isomerization towards an alcohol or an electron-withdrawing group were successfully developed[37–40]. We, therefore, anticipated that the migration of a secondary organopalladium along the chain by fast β-hydride elimination and subsequent re-addition could eventually be driven by the ring-strain release from a Pd[II]-catalysed β-cyclopropyl fragmentation[41]. In case of a selective C–C bond cleavage, a polysubstituted stereodefined cyclopropane—whose synthesis and reactivity with transition metals have been well established through diverse strategies[42–45] would ultimately afford the acyclic internal olefin bearing the quaternary carbon stereocentre (Fig. 2c)[46–48]. Our proposed sequence stems from a detailed analysis of the potential factors driving the reaction to completion but also considers the ease of preparation of stereomerically enriched starting materials (**1a–1s**, dr ≥ 92:08:0:0) (ref. 45). To start with, following the oxidative addition of an aryl halide to $L_nPd(0)$ complex, two regioisomers could result from the 1,2-aryl migratory insertion of the terminal olefin into the Pd(Ar) bond. Both resulting organopalladium species **I** and **II** could suffer β-hydride elimination and $L_nPdHX$ could either disengage from the molecule to afford the alkenyl derivative, or undergo readdition of the hydride on the isomerized olefin to trigger the chain-walking. Assuming that the palladium-walk would be an efficient process, it would reach at a certain point the sterically congested cyclopropane (**III** and **IV**, respectively) where a C–C bond cleavage might occur at its β-position. Both C–C bonds ($C_1$–$C_2$ or $C_1$–$C_3$) could presumably be cleaved, but intermediates **V** and **VI** resulting from a selective $C_1$–$C_2$ bond cleavage would enable further chain-walking towards the alcohol. One additional question would concern the configurational stability of the resulting tertiary organopalladium intermediate in **V** and **VI** as well as the stereofacial choice of the migration event. The continuation of the Pd-walk would ultimately end with the disengagement of $L_nPdHX$ to extrude the expected carbonyl derivative (**2** and/or **3**), and, regeneration of the active catalytic species $L_nPd(0)$ by reaction of the palladium hydride with a base. Therefore, the success of this rather convoluted intramolecular process would reckon on a complete control of all of these numerous elementary steps.

In lights of these elements, the selective conversion of **1** into **2** and/or **3** might appear strenuous, considering the possible pitfalls and potential by-products.

**Arylation of allylcyclopropyl carbinol derivatives**. Using our model substrate, namely the syn allyl cyclopropyl carbinol ( ± )-**1a** ($R^1$ = Bu, $R^2$ = Me, $R^3$ = $R^4$ = H, n = m = 0), synthesized diastereomerically enriched[45], we first determined the optimal conditions for our proposed sequence using phenyl iodide for a Pd-based catalytic system. We found that in the presence of a catalytic amount of Pd(OAc)$_2$ (5 mol%), *tris*(4-trifluoromethyl-phenyl)phosphine as ligand (15 mol%), NaHCO$_3$ as inorganic

**Figure 1 | Comparison between the different strategies for the functionalization of organic skeletons.** (**a**) Classical approach for the selective interconversion of a functional group. (**b**) Chemoselective interaction between a metal and a specific moiety allowing the creation of functionalized organometallic species, leading to a variety of electrophilic substitution reactions. (**c**) Remote functionalization through metal migration strategy enabling a transformation away from a functional group at a distant position (FG²); located further away from (FG¹). Despite many inherent advantages, this approach does not allow the presence of the highly regarded quaternary carbon stereocentre. (**d**) Proposed sequence where a strategically stereodefined polysubstituted cyclopropane vault is used assuming that the ring-strain release may drive the chain-walking process towards the formation of acyclic products and allowing the introduction of an internal substitution on the carbon skeleton. (**e**) Illustration of this sequence was reported for the addition of stoichiometric amount of low-valent zirconocene derivatives to ω-alkenyl cyclopropanes generating a cascade of events composed by successive allylic C–H bond activations, selective C–C bond cleavage and finally reaction with two different electrophiles (E¹X and E²X).

base (2.5 equiv), tert-butylammonium chloride (2 equiv) and molecular sieves 4 Å (150 mg mmol⁻¹ substrate), **1a** and PhI (1.2 equiv) in tetrahydrofuran (THF) at 95 °C for 24 h afforded a mixture of aldehydes **2a** and **3a** in 73% combined yield with good regioselectivity (10:1) and as unique (*E*)-isomers (Fig. 3b, for the optimization table see Supplementary Table 1). No other positional alkene isomers could be detected suggesting that not only the final product was at no risk of further olefin isomerization but also that the chain-walking process was exclusively triggered by the initial Heck arylation. Following this optimization based on the coupling of phenyl iodide with **1a**, other aryl halides were tested (Fig. 3b). To our delight, aryl bromides were also suitable partners as no significant difference were observed in the chemical outcome of the reaction (see Fig. 3b, formation of **2a:3a**). This transformation was compatible

with the presence of functional groups on the aryl moiety such as halides (**2d**–**2g**), electron-donating group (**2b** and **2c**) and electron-withdrawing group including an interesting methyl ketone moiety (**2h**). Notably, 2-bromothiophene was also successfully incorporated onto **1a**, albeit in lower yield and regioselectivity (**2i**). Not only the aryl moiety could be varied but the substitution pattern on the quaternary stereocentre (R¹ and R²) may as well be easily modified through our previously described flexible approach for the synthesis of such cyclopropane rings[43,45] (Fig. 3c). Indeed, compounds bearing the quaternary carbon stereocentre possessing linear (Me, Et, Bu, CH₂CH₂OBn), branched (*i*Bu), cyclic (*c*Hex) and aromatic (Ph) groups could be synthesized using similar strategy (**1b**–**1g**). We observed only few differences in the chemical outcome of the Heck coupling domino reaction, as the different starting materials all afforded

**Figure 2 | Examples of Pd-catalysed Heck arylation as an initiation step triggering a remote functionalization and mechanistic hypothesis.** (**a**) The first report of a Pd-intramolecular migration across a molecule was pioneered by Heck[35] and later extended by Larock[36] showing the versatility of the method across longer alkenyl chain. (**b**) Sigman reported the enantioselective and site-selective Heck reaction of diazonium salts, vinyl triflates, aryl boronic acids and indole derivatives with ω-alkenyl alcohols[13,37–40]. (**c**) Proposed catalytic sequence and mechanistic hypotheses for the transformation of cyclopropane 1 into regioisomers 2 and/or 3.

the corresponding aldehydes (**2j**–**2o**) in good yields (59–81%) and regioselectivities (>5.7:1). It should be emphasized that our method could potentially generate carbon quaternary stereocentres of rather similar steric groups, such as butyl and iso-butyl chains, on an acyclic system (**2k**). Additionally, secondary alcohol derivatives ($R^4 = $Me, **1h** and **1i**) also underwent similar

transformation to mainly yield the corresponding methyl ketones **2p**–**2t** (Fig. 3d).

**Variation of chain-length in substrates**. Most importantly, the exact same conditions still proceeded on analogous substrates

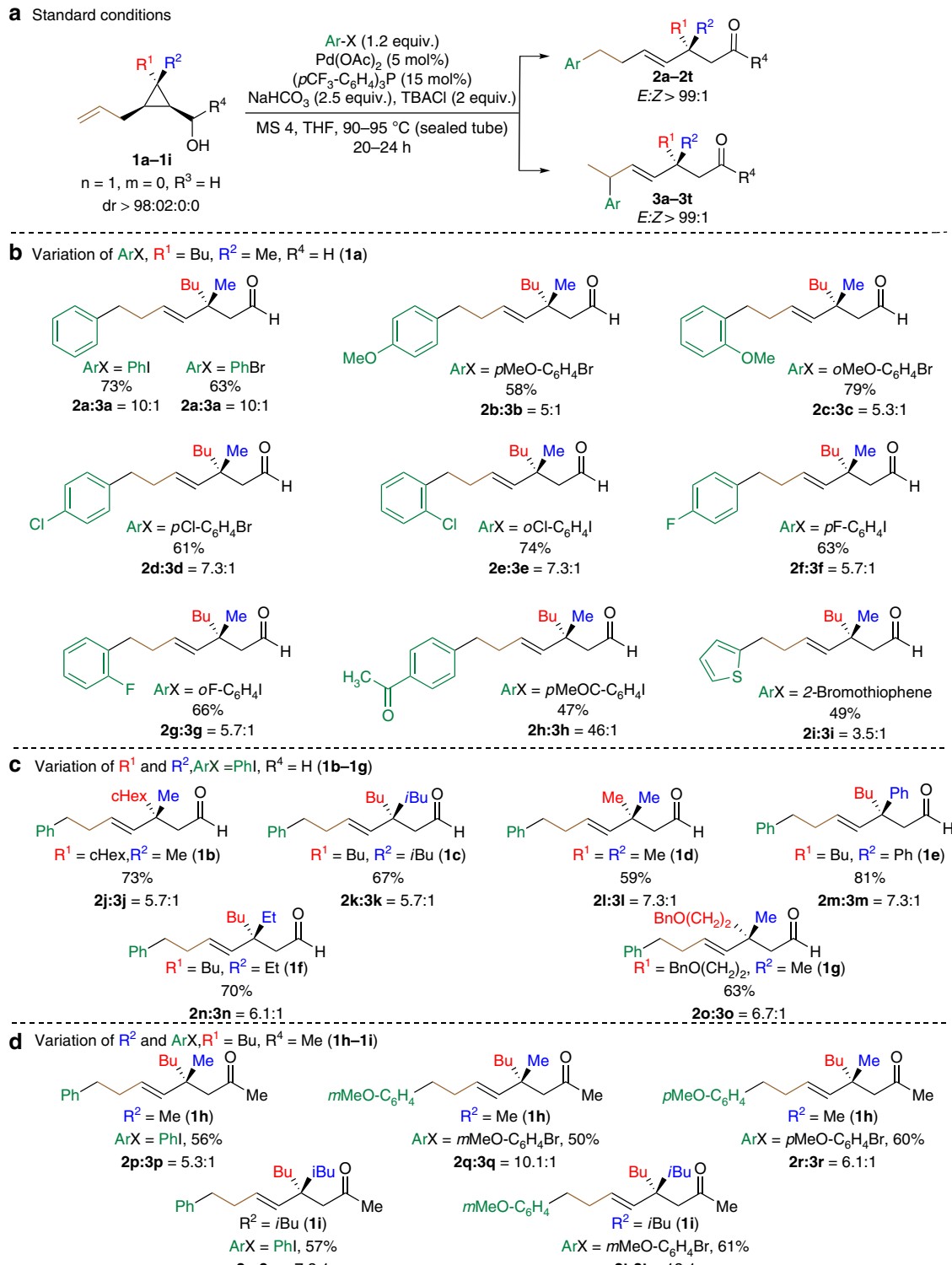

**Figure 3 | Scope of the reaction by varying ArX, R$^1$, R$^2$ and R$^4$.** (**a**) Standard experiment conditions for the coupling of **1** with ArX yielding **2** and **3** using a Pd(0)-based catalytic system. (**b**) Scope of the aryl halide (ArX). (**c**) Scope of the substitution pattern on the cyclopropyl carbinol moiety (R$^1$ and R$^2$). (**d**) Examples involving the transformation of secondary alcohol derivatives (R$^4$ = Me) into ketones. Regioisomeric and E/Z ratios were determined by NMR and GC analyses of the crude reaction mixture. All represented yields are the combined yields of both regioisomers after purification by column chromatography.

with different chain-lengths (*n* and *m*), further demonstrating the efficient palladium chain-walking mechanism (Fig. 4, for the sake of clarity, all the following structures of the minor products **3u–3ap** are omitted). The examples shown in Fig. 4, illustrates the case where 1 to 5 methylene units were present between the unsaturation and the cyclopropyl ring (**2u–2ac**). In parallel, increasing the molecular distance between the alcohol moiety and the cyclopropane (*m*) also led to acceptable yields of the

corresponding carbonyl derivatives (**2ad–2af**, Fig. 4). Lengthening simultaneously both sides (**1o**, $n = 4$, $m = 1$) did not prevent the reaction to occur (**2ag**, Fig. 4), demonstrating the high versatility of the method. In all cases, as already previously mentioned, the $E/Z$ ratio of the major 1,2-disubstituted olefin obtained in these products remained excellent ($E/Z > 99:01$).

Encouraged by these positive results, we added one more level of complexity to this sequence with the unfolding of 1,2,2,3,3-pentasubstituted cyclopropanes as illustrated in Fig. 5. In this particular case, the selective C–C bond cleavage could be challenging as both substituted distal C–C bonds ($C_1$–$C_2$ versus $C_1$–$C_3$) have a similar substitution degree. To our delight, when **1p** was treated in our optimized experimental conditions with two different ArX, only the products **2ah** and **2ai** resulting from a selective ring-cleavage were obtained in good yields as described in Fig. 5 suggesting the primordial role of the alcohol moiety in the selective C–C bond cleavage. However, the benzylic stereogenic centre was obtained as two diastereoisomers in an almost 1:1 ratio, presumably owing to an epimerization during the last

step of the reaction sequence namely the keto-enol tautomerization as would be probed in our isotope labelling investigations discussed in Fig. 6. If our hypothesis that the keto-enol tautomerism induced epimerization of the stereocentre were true, the reaction on 1,1,2,2,3-pentasubstituted cyclopropanes possessing the alcohol moiety at least one carbon away of the tertiary stereocentre would be insensitive to the last chemical event (keto-enol tautomerism) and the stereochemistry of the tertiary stereocentre would, therefore, be preserved throughout the process.

Indeed, the remote arylation of ω-alkenyl cyclopropyl alcohols **1q–1t** ($m \geq 1$) were tested with various ArX and we were delighted to see that compounds **2aj–2ap** were obtained in satisfactory yields as a single set of diastereoisomers and as a single $E$-isomer as observed by NMR and GC analyses ($dr = 98:02$) suggesting that the Pd does not disengage during the process and migrates on the same stereoface. The relative configuration was confirmed by X-ray analysis of the hydrazone derivative (**4**) of **2aj** (see Supplementary Fig. 91). It should be

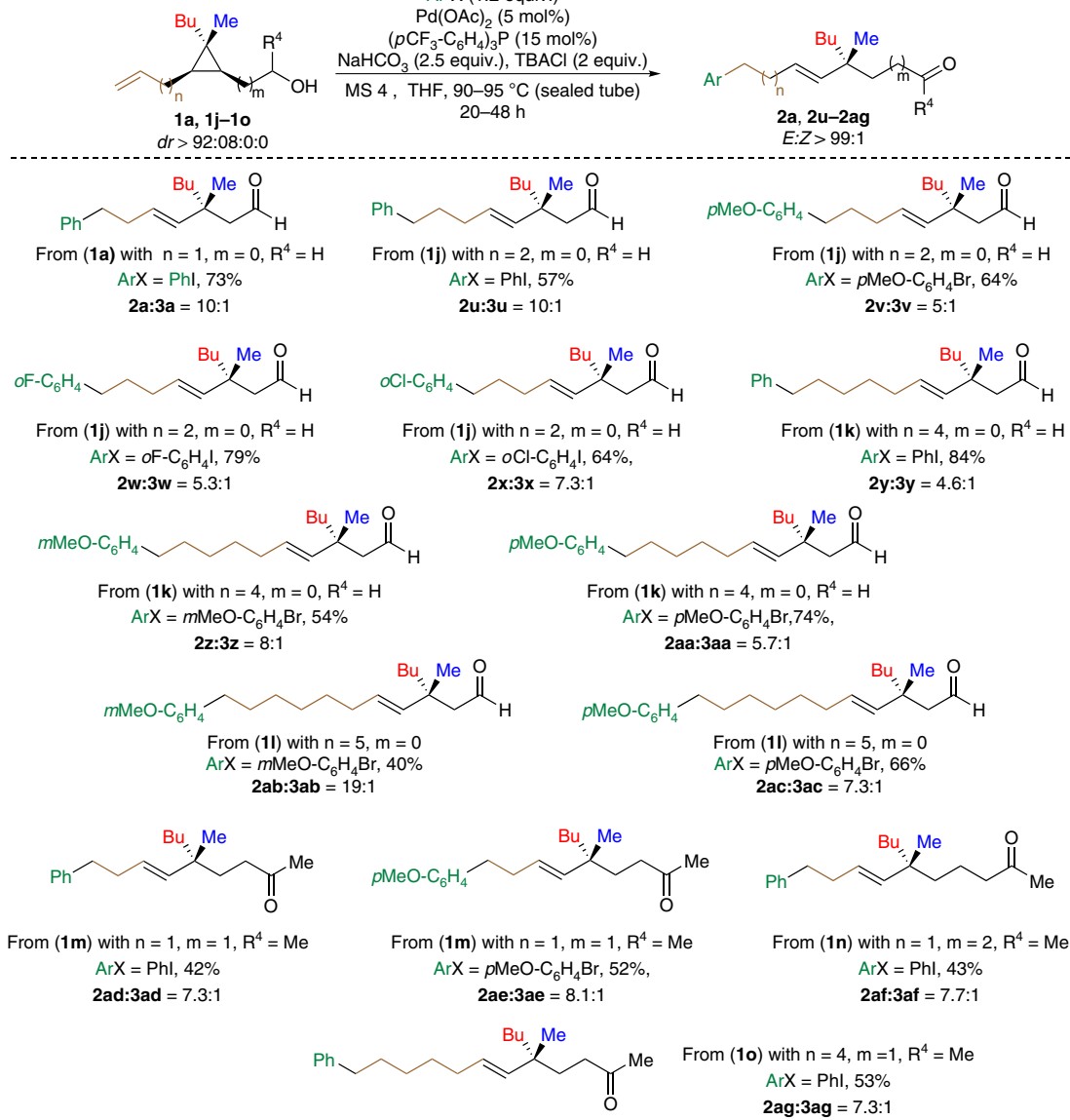

**Figure 4 | Scope of the chain lengths (*n* and *m*).** Increase of the molecular distance (*n*) between the double bond and cyclopropane moieties as well as the molecular distance (*m*) between the alcohol and cyclopropane and finally the situation where both chain length *n* and *m* are increased.

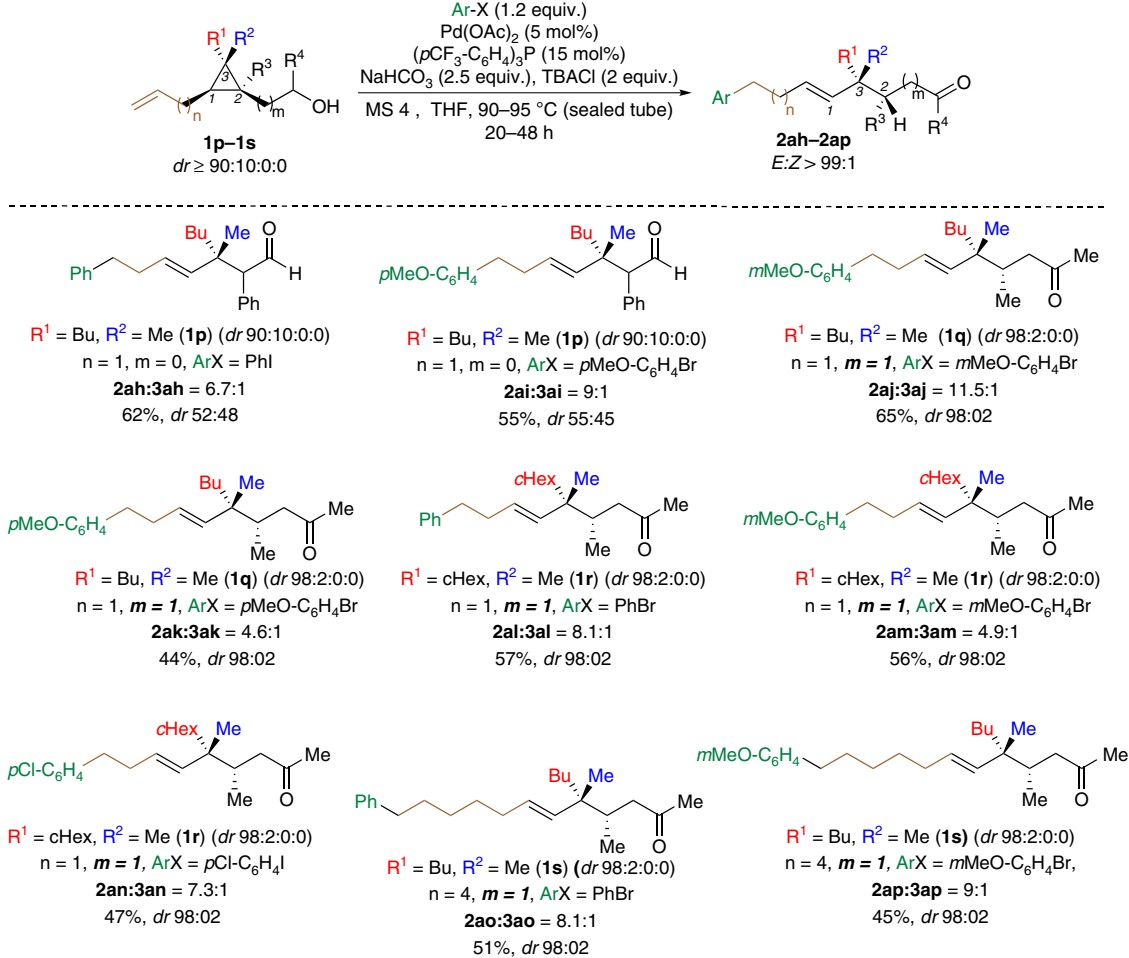

**Figure 5 | Scope of the chain lengths (n and m) and substitution.** Remote Pd-catalysed Heck triggered ring-opening in 1,2,2,3,3-pentasubstituted cyclopropane derivatives (**1p**–**1s**). The relative configuration of **2aj** was confirmed by X-ray analysis of the corresponding hydrazone **4** (see Supplementary Fig. 91). Relative configurations of other major products **2aj**–**2ap** were assigned by analogy. The described yields are the combined yields of both regioisomers after purification by column chromatography. Regioisomeric, diasteromeric and E/Z ratios were determined by NMR and GC analyses of the crude mixture.

emphasized that the formation of two contiguous stereocentres, including the formation of a quaternary carbon stereocentre, is still a challenging issue in synthesis and this approach proposes a highly stereoselective alternative route to such molecular backbones.

**Mechanistic insights.** To probe our mechanistic hypotheses, we conducted a series of control, including isotope labelling, experiments. We first studied the effect of the relative stereochemistry between the ω-alkenyl chain and the alcohol residue with PhI as coupling partner (Fig. 6a). When the *anti* diastereoisomer ( ± )-**1a**$_{anti}$ (dr = 90:10:0:0) was treated under the same experimental condition as ( ± )-**1a**$_{syn}$, the same final aldehyde **2a** was obtained in similar yields and E/Z-selectivity albeit in lower **2a:3a** isomeric ratio (**3a** not indicated in the Fig. 6). Isotope labelling experiments were also performed to map the metal-complex migration across the organic molecule (Fig. 6b). When two geminal deuterium atoms were located either at the terminal double bond or at the homoallylic positions (**5** and **6**, respectively), the expected products **11** and **12** were independently obtained confirming the well-established Pd-catalysed 1,2-hydrogen shift mechanism. Positioning tactically a deuterium atom on the cyclopropyl ring as illustrated in **7** enabled us to suggest that the ring-opening should proceed faster than a

potential β-hydride elimination that would have afforded an alkylidenecyclopropane intermediate[49]. When a deuterium atom was strategically introduced at the other cyclopropyl carbon atom as in **8**, the product **14** obtained after our cascade reaction was deuterium atom-free. A similar result was observed when **9** was treated under our Heck conditions as the monodeuterated aldehyde **15** was selectively obtained without traces of deuterium in α-position to the carbonyl group. These losses of deuterium atoms during these two keto-enol tautomerizations suggested a rapid exchange that was also accounted for the epimerization of the stereogenic centre in **2ah** and **2ai** as previously discussed (Fig. 5). However, when the keto-enol tautomerization took place one carbon further away owing to the introduction of one additional carbon centre as in **10**, the reaction proceeded with preservation of the deuterium atom and **16** was obtained as a single diastereoisomer.

All these experiments enabled us to corroborate the proposed mechanism presented on Fig. 2 where the oxidative insertion of the aryl halide to L$_n$Pd(0) led to the LnPd(II)ArX species from which the migratory insertion on **1** occurred to deliver **II**. A facile and rapid migration of the metal over the alkyl chain through successive 1,2-hydrogen shifts (chain-walking) then leads to the cyclopropyl methyl palladium species **III**. At this stage, a selective ring-fragmentation proceeds into the unique formation of

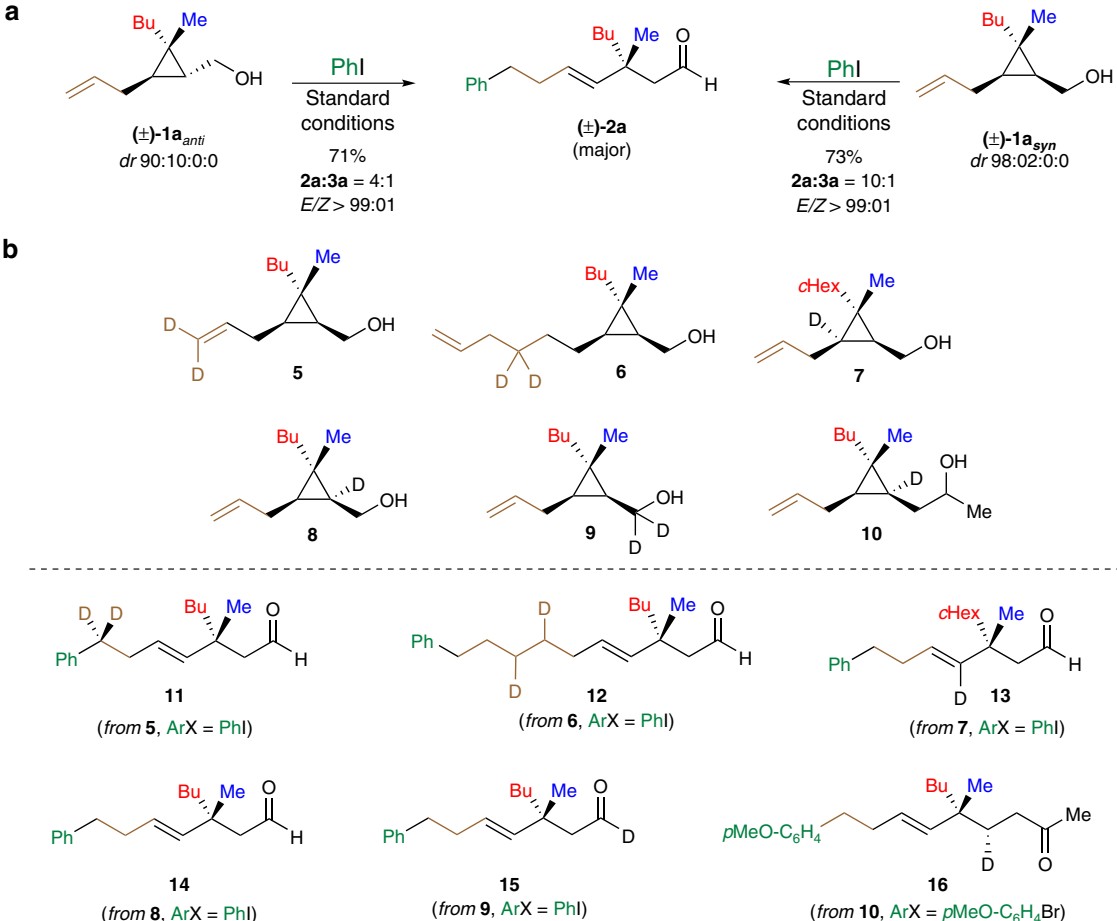

**Figure 6 | Control and isotope labelling experiments.** (**a**) Effect of the relative stereochemistry between the ω-alkenyl chain and the alcohol residue on the reaction outcome using standard reaction conditions (see Fig. 3a). (**b**) Isotopic labelling experiments. Deuterium incorporation was confirmed by NMR and mass spectrometry analyses of products **5**–**16** after purification over column chromatography.

(*E*)-homoallyl organopalladium **V**. The highly selective C–C bond cleavage (when $R^3 \neq H$) might be rationalized by the directionality of the second chain-walking event. Furthermore, the resulting tertiary organopalladium species ($R^3 = Me$) did not suffer any epimerization when $m \geq 1$ and continue to promote the chain walking on the same stereoface to finally yield the desired carbonyl derivative **2aj**–**2ap** as the major products. When $m = 0$ and, therefore, no additional chain-walking process can be performed, and the keto-enol tautomerization causes the epimerize of the stereocentre.

## Discussion

In conclusion, we could report that a single transition metal based-catalytic system could successively mediate a (i) regioselective arylation of the terminal olefin, (ii) a chain-walking, (iii) a selective C–C bond cleavage of the cyclopropane, (iv) additional chain-walking and (v) release the carbonyl moiety to regenerate the active catalytic species. All these consecutive elementary steps proceeded in a one-pot operation with impressive levels of selectivities. During this study, we could uncover the highly selective C–C bond cleavage conditioned by the chain-walking directionality. As a result, the strategically positioning of the cyclopropane 'vault' enabled the consecutive introduction of a tertiary and a quaternary stereocentre at unrelated positions between two distant functional groups into a linear hydrocarbon

skeleton. This contrasts with catalytic chain-walking reactions that are generally perceived as proceeding exclusively across methylene chains. Additionally, the ease of synthesis of such diastereomerically pure ω-ene cyclopropyl derivatives (see Supplementary methods) associated with this proposed versatile and mild procedure allowed the downstream functionalization into further sophisticated acyclic hydrocarbon frameworks.

## Methods

**Materials.** For NMR spectra of compounds in this manuscript, see Supplementary Figs 1–90. For the optimization of reaction conditions of compound **2a**, see Supplementary Table 1. For the crystallographic data of compounds **4**, see Supplementary Fig. 91, Supplementary Table 2 and Supplementary Note 1. For the experimental procedures and analytic data of compounds synthesized, see Supplementary Methods.

**Preparation of compounds 2 and 11–16.** A sealed vial (10 ml capacity) equipped with a magnetic stirring bar, was successively charged with an alcohol derivative **1a-1 s/5-10** (0.2 mmol), aryl bromide/iodide (0.24 mmol), Pd(OAc)₂ (2.24 mg, 5 mol%), ($p$CF₃-C₆H₄)₃P (14 mg, 15 mol%), NaHCO₃ (42 mg, 2.5 mmol), pre-activated molecular sieves (4 Å size, powdered, 30 mg) and tetrabutylammonium chloride (2 mmol) followed by addition of dry THF (0.4 ml). After sealing, the reaction mixture was stirred for 10 min at room temperature to obtain a homogeneous solution and was then placed on a pre-heated oil bath at 90–95 °C for 24–48 h. After completion of the reaction (as monitored by TLC), the reaction mixture was cooled down to room temperature, diluted with Et₂O (5 ml) and filtered. The obtained residue was concentrated and subjected to a column chromatography using 3–8% Et₂O in hexane as eluent to afford product **2a–2z**,

**2aa–2ap** and **11–16** as a mixture with their corresponding inseparable regioisomers **3a–3z**, **3aa–3ap** and **11\*-16\***.

**Data availability.** The X-ray crystallographic coordinates for structure of compound **4** reported in this article have been deposited at the Cambridge Crystallographic Data Centre (CCDC) with the accession code CCDC 1500847. This data can be obtained free of charge from the Cambridge Crystallographic Data Centre via www.ccdc.cam.ac.uk/data_request/cif. The authors declare that all other data supporting the findings of this study are available within the article and Supplementary Information files, and also are available from the corresponding author upon reasonable request.

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

## Acknowledgements

This research was supported by the European Research Council under the Seventh Framework Program of the European Community (ERC grant agreement no. 338912). S.S. thanks the Planning and Budgeting Committee (PBC) of the Council for Higher Education for providing financial support. I.M. holds the Sir Michael and Lady Sobell Academic Chair.

## Author contributions

S.S., J.B. and A.V. planned, conducted and analysed experiments. I.M. designed and directed the project and wrote the manuscript with contributions by S.S., J.B. and A.V. All authors contributed to discussions.

## Additional information

**Competing financial interests:** The authors declare no competing financial interests.

