## [Peer review file · Nature Communications]

REVIEWERS' COMMENTS:

Reviewer #1 (Remarks to the Author):

The communication by Marek and co-workers describes the development of a Pd-catalyzed Heck arylation of alkenyl alcohols containing an intercalated cyclopropane motif. This approach enables to construct a novel C-C bond with high site-selectivity while the in situ generated Pd-hydride 'walks' along the carbon skeleton, selectively ring-opens the cyclopropane and refunctionalizes the terminal alcohol functionality into a carbonyl. Starting from diastereomerically pure cyclopropyl-containing alkenyl alcohols, strategic positioning of the cyclopropane motif into the linear alkyl chain of the substrates permits to generate adjacent tertiary and quaternary stereocenters. The scope of the method is broad from both coupling partners and the dr obtained range from practical to synthetically useful. The products obtained would be difficult to access by alternative approaches. Convincing and elegant isotopic labelling experiments support the mechanistic hypotheses (i.e. 'chain-walking') initially proposed by the authors.

From a broader perspective, this study is highly original and falls within the emerging field of remote functionalization using atom-economical and selective organometallic catalysis. As clearly and rightfully set in the introduction by the authors, this approach has the potential to change one may conceptualize synthesis and retro-synthesis in the coming years.

In conclusion, not only I do strongly support publication of this fine piece of work in Nature Communication, but I also strongly encourage the Editors to consider Nature Chemistry as a more appropriate forum for its dissemination.

Reviewer #2 (Remarks to the Author):

The manuscript by Marek and co-workers describes a Pd-catalyzed cyclopropane opening reaction triggered by a Heck-type arylation of a remote terminal olefin. Established by Heck and Larock, the Heck arylation followed by olefin migration via a metal-catalyzed beta-hydride elimination/re-addition mechanism has been known for decades to functionalize remote positions of an olefin substrate. However, one major drawback of such a classical reaction is the lack of compatibility of a quaternary carbon center inside the chain. In the present work by Marek, such a problem is circumvented by incorporation of a Pd-catalyzed cyclopropane ring opening. While from a mechanistic viewpoint such a transformation may not be that novel, especially when considering the well precedence of each unit of the

transformation in literature, it is nevertheless synthetically useful from a practicality viewpoint. First, the stereochemistry of the cyclopropane moiety, which can be readily controlled during substrate synthesis, can be completely transferred to the open chain product. Second, in the case of 1,1,2,2,3-pentasubstituted cyclopropanes, the consecutive introduction of a quaternary and a tertiary stereocenter into a linear skeleton is also impressive. In addition, the mechanistic study through D-labelling is convincing. Thus, this reviewer favors its publication in Nature Communications after the following minor issue is addressed.

While a panel of functional groups on the aryl part as well as different substitution patterns on the cyclopropane motif are tolerated, it would be great to show 1) if this reaction can also tolerate cyclopropanes with only one substituent on the C1 position (e.g. R1 = Bu, R2 = H), and 2) if this reaction can use an EWG (such as an ester group) instead of an alcohol as a terminus to access α,β -unsaturated systems.

Reviewers Comments

Reviewer #1 (Remarks to the Author):

The communication by Marek and co-workers describes the development of a Pd-catalyzed Heck arylation of alkenyl alcohols containing an intercalated cyclopropane motif. This approach enables to construct a novel C-C bond with high site-selectivity while the in situ generated Pd-hydride 'walks' along the carbon skeleton, selectively ring-opens the cyclopropane and refunctionalizes the terminal alcohol functionality into a carbonyl. Starting from diastereomerically pure cyclopropyl-containing alkenyl alcohols, strategic positioning of the cyclopropane motif into the linear alkyl chain of the substrates permits to generate adjacent tertiary and quaternary stereocenters. The scope of the method is broad from both coupling partners and the dr obtained range from practical to synthetically useful. The products obtained would be difficult to access by alternative approaches. Convincing and elegant isotopic labelling experiments support the mechanistic hypotheses (i.e 'chain-walking') initially proposed by the authors.

From a broader perspective, this study is highly original and falls within the emerging field of remote functionalization using atom-economical and selective organometallic catalysis. As clearly and rightfully set in the introduction by the authors, this approach has the potential to change one may conceptualize synthesis and retro-synthesis in the coming years.

In conclusion, not only I do strongly support publication of this fine piece of work in Nature Communication, but I also strongly encourage the Editors to consider Nature Chemistry as a more appropriate forum for its dissemination.

Answer from the authors: No additional comments

Reviewer #2 (Remarks to the Author):

The manuscript by Marek and co-workers describes a Pd-catalyzed cyclopropane opening reaction triggered by a Heck-type arylation of a remote terminal olefin. Established by Heck and Larock, the Heck arylation followed by olefin migration via a metal-catalyzed beta-hydride elimination/re-addition mechanism has been known for decades to functionalize remote positions of an olefin substrate. However, one major drawback of such a classical reaction is the lack of compatibility of a quaternary carbon center inside the chain. In the present work by Marek, such a problem is circumvented by incorporation of a Pd-catalyzed cyclopropane ring opening. While from a mechanistic viewpoint such a transformation may not be that novel, especially when considering the well precedence of each unit of the transformation in literature, it is nevertheless synthetically useful from a practicality viewpoint. First, the stereochemistry of the cyclopropane moiety, which can be readily controlled during substrate synthesis, can be completely transferred to the open chain product. Second, in the case of 1,1,2,2,3-pentasubstituted cyclopropanes, the consecutive introduction of a quaternary and a tertiary stereocenter into a linear skeleton is also impressive. In addition, the mechanistic study through D-labelling is convincing. Thus, this reviewer favors its publication in Nature Communications after the following minor issue is addressed.

While a panel of functional groups on the aryl part as well as different substitution patterns on the cyclopropane motif are tolerated, it would be great to show 1) if this reaction can also tolerate cyclopropanes with only one substituent on the C1 position (e.g. R1 = Bu, R2 = H), and 2) if this reaction can use an EWG (such as an ester group) instead of an alcohol as a terminus to access alpha,beta-unsaturated systems.

Answer from the authors:

- 1) The reaction can indeed tolerate only one substituent of the C1 position of the three-membered rings albeit in lower overall yield. This particular transformation has not been highlighted in this manuscript since the formation of a single tertiary stereocenter could be achieved using the classical palladium-walk through a tertiary stereocenters as shown by the pioneering work of Matthew Sigman in a much better yield. Therefore, we considered that the novelty of such particular transformation would be of much less of interest to the scientific community.
- 2) The reaction with an electron-withdrawing group directly connected to the cyclopropane has not been attempted as the ring-opening would provide a Pd-enolate that would not be able to be regenerated through a beta-H elimination reaction. A completely different termination system would then be necessary and was not the purpose of this work. On the other hand, if one introduces the ester one more carbon away from the strained ring (i.e. cyclopropyl-CH₂-COOEt), then the access to alpha-beta unsaturated systems would be realizable and certainly be of synthetic interests. This suggestion has not yet been performed but will be surely considered for any upcoming full-paper.